# Null-Result Test for Effect on Weight from Large Electrostatic Charge

**George Hathaway [1] and Lance L. Williams [2],\***

1   Hathaway Research International, Toronto, ON L5T 1S1, Canada; george@hathawayresearch.com
2   Konfluence Research Institute, Manitou Springs, CO 80829, USA
\*   Correspondence: willi@konfluence.org; Tel.: +1-719-685-1163

**Abstract:** We report test results searching for an effect of electrostatic charge on weight. For conducting test objects of mass of order 1 kg, we found no effect on weight, for potentials ranging from 10 V to 200 kV, corresponding to charge states ranging from $10^{-9}$ to over $10^{-5}$ coulombs, and for both polarities, to within a measurement precision of 2 g. While such a result may not be unexpected, this is the first unipolar, high-voltage, meter-scale, static test for electro-gravitic effects reported in the literature. Our investigation was motivated by the search for possible coupling to a long-range scalar field that could surround the planet, yet go otherwise undetected. The large buoyancy force predicted within the classical Kaluza theory involving a long-range scalar field is falsified by our results, and this appears to be the first such experimental test of the classical Kaluza theory in the weak field regime, where it was otherwise thought identical with known physics. A parameterization is suggested to organize the variety of electro-gravitic experiment designs.

**Keywords:** gravity; electrostatics; experiment

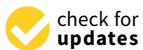

## 1. Introduction

This paper reports the results of a search for possible weight changes on electrically charged objects. Although our test is motivated by a search for a long-range scalar field, our work appears to be among the first reported tests for an electrical influence on gravity undertaken for a wide range of charge states.

The earliest tests for coupling between gravity and electromagnetic forces were measurements by Faraday [1]. His diary for 19 March 1849 notes:

> Gravity. Surely this force must be capable of an experimental relation to Electricity, Magnetism and the other forces, so as to bind it up with them in reciprocal action and equivalent effect.

One experiment involved dropping a helical wire to ascertain whether a current was induced in the fall. He coined the term "gravelectric current" for such an effect, but did not find it, or any other [2] .

Over approximately two decades, from the 1930s into the 1950s, a series of "electro-gravitic" force patents was assigned to T.T. Brown [3–5]. Brown claimed to discover, serendipitously, that a charged capacitor experiences a force. Brown's early tests were on capacitors banked in a stack, suspended horizontally, so the capacitor plate surface normals were horizontal. The resulting claimed force from this configuration was horizontal. Brown refined his design over time, with the "essential teaching" captured in his patent from 1965 [6], which settled on the necessity of asymmetric capacitors, where one electrode is very large and one very small. In 1968, Brown defined his electro-gravitic force as "the pondermotive force developed within a high-K dielectric under electrical strain". He goes on to describe it as "independent of the orientation with respect to the Earth's gravity vector [7]."

None of Brown's patents had any known basis in physics or engineering, and he offered no equations to describe the principles underlying his designs. It is presumed he found his effects empirically, without a coherent mathematical framework to predict or describe the phenomena, as we otherwise have come to expect from mathematical laws of physics. Brown acknowledged in 1968 that none of his devices ever lifted from the ground [7]. The source of momentum in Brown's results is now understood to be due to coronal wind induced in the atmosphere around charged objects. The effect vanishes when the atmospheric density goes to zero, implying there is no new physics, and the phenomenon is purely electric, arising from the known properties of gases exposed to strong electric fields [8–10].

Electrically-charged torque pendulums were investigated, starting in 1964, by Saxl [11]. Saxl claimed to see an alteration in the period of a torque pendulum when it was electrically charged. Saxl further claimed some correlation to environmental phenomena, and he suggested some utility for geophysical measurement. Subsequent researchers have not reproduced an effect on a charged pendulum, and it is considered a null result [12,13].

Recently, Tajmar and Schreiber [10] tested for weight changes on a variety of small capacitors held in different spatial configurations. These capacitors were tens of millimeters in size, and had plate distances of a few millimeters. They were tested up to 10 kV, and no weight variations were seen in any tests.

The effects being probed by the experiments described above might be expected from a variety of possible physical mechanisms. Tajmar and Schrieber describe some of the underlying theories implicated in their tests, and yet others exist. In fact, there are at least three different fundamental mechanisms that could be at work in a force test on a charged body. Such an effect could be attributed to an effect on the gravitational mass, an effect on the inertial mass, or a new force altogether. The literature does not make clear how one is to interpret or inter-relate the variety of experiments in this area. Therefore, let us develop these theoretical concepts before describing the experiment.

The advent of general relativity revealed a coupling between electromagnetism and gravity in that any source of energy-momentum, even electromagnetic energy-momentum, is a source of gravity. Therefore, electromagnetic energy can conceivably create gravitational fields. Yet, electromagnetic energy densities are so low compared to the energy density of condensed matter that electromagnetic sources of gravity are insignificant. Conversely, gravity famously affects electromagnetic fields by deflecting light rays, and this effect can be significant.

Surprising complexity can arise in the gravito-electromagnetic interaction described by general relativity. The Maxwell equations in curved spacetime famously take on complex effects. And the conventional Reissner–Nordstrom metric for a source of mass $M$ and electric charge $Q$ provides an intriguing effect: at a certain critical radius $r_c \propto \mu_0 Q^2 / M$, where $\mu_0$ is the permeability of free space, the gravitational field vanishes due to an electrostatic effect working against gravity. It is remarkable that the electric field causes gravity to vanish even for uncharged bodies. Electrostatic field energy enters the Reissner–Nordstrom metric with a sign opposite to that expected from the mass-energy of the Schwarzschild metric.

The classical Kaluza theory elegantly unifies a new long-range scalar field with the long-range vector field of electromagnetism, and the long-range tensor field of general relativity. A recent calculation shows that the classical Kaluza theory predicts a strong electro-gravitic buoyancy force for electrically-charged objects from the planetary scalar field [14]. The long-range scalar field effects are distinct from either a variation in inertial mass or gravitational mass, and constitute instead a force of nature undetected so far. Since there had been little testing of such effects in the literature, we undertook the test campaign reported here. Our test is the first unipolar, high-voltage, meter-scale, static test for electro-gravitic effects. The theory detail and parameterization of the Kaluza buoyancy force is given in the next section.

There are clearly a variety of experiments besides our own that purport to test anomalous couplings between gravity and electromagnetism, so we would like to suggest a parameterization inspired by the Kaluza theory. We can parameterize all electro-gravitic experiments with the dimensionless ratio of charge to mass, $(Q/M)/\sqrt{4\pi G \epsilon_0}$, where $G$ is the Newtonian gravitational constant and $\epsilon_0$ is the permittivity of free space. The value of the Kaluza scalar field buoyancy force calculated in [14] is proportional to the square of the dimensionless ratio. This ratio also emerges in the ADM mass [15]. Such a parameterization can help to avoid re-verifying the same parameter regime in different configurations.

It is useful to convert between charge and voltage. Electric charge is the quantity that enters the scalar force expression, not voltage. They are related by the capacitance $C = Q/V$. The capacitance of our test articles is approximately $4\pi\epsilon_0 R$, where $R$ is the effective radius of the test object. For our test articles, the capacitances are approximately $10^{-10}$ mks. Therefore, at 200 kV, the charge is about $2 \times 10^{-5}$ coulombs.

We tabulate the dimensionless charge-to-mass ratio for each test, along with the weight measurements, to help contextualize the parts of electro-gravitic parameter space we have explored. We tested a range of values spanning four orders of magnitude, from 1 to $10^4$. By comparison, the torsion pendulum tests by the authors of [12,13] were all at approximately 3000. The button capacitor tests by the authors in [10] were all at approximately $10^{10}$. A Millikan oil drop of size one micron and carrying one free electron would have a dimensionless charge-to-mass ratio of approximately $10^5$. The recent test for electrostatic effects on time dilation [16] were at approximately $10^4$. These parameterizations are summarized in Table 1.

**Table 1.** Parameterization of various electro-gravitic experiments, according to the dimensionless parameter $(Q/M)/\sqrt{4\pi\epsilon_0 G}$, where $Q$ is electric charge obtained in the experiment, and $M$ is mass of the charged object.

| Electro-Gravitic Experiment | $(Q/M)/\sqrt{4\pi\epsilon_0 G}$ |
|---|---|
| large horizontal capacitor plate on a torque pendulum [12,13] | 3000 |
| button-size capacitor weight measurement [10] | $10^{10}$ |
| unipolar, electrically-charged sphere weight measurement (this work) | 1 to $10^4$ |
| clock in electrically-charged enclosure [16] | $10^4$ |
| Millikan oil drop, 1 micron in size with 1 electron | $10^5$ |

## 2. Kaluza Buoyancy Force Detail

Our unipolar, high-voltage, meter-scale, static test for electro-gravitic effects is among the first such reported in the literature. While our experimental results may be of interest to a range of theories coupling gravity and electromagnetism, we were motivated by one theory in particular, and to test one prediction in particular. That is the electro-gravitic buoyancy calculated in [14] from the Kaluza theory.

The year 2021 is the centennial anniversary of the Kaluza theory, and it is not our purpose here to summarize 100 years of research into that fascinating theory. Our purpose is to provide an experimental test for a prediction made in [14], and we refer the reader to that paper for comprehensive historical references. Nonetheless, it is useful to briefly contextualize the Kaluza theory treated in [14], and to distinguish it from other similar variants, before going on to the quantitative parameterization of the buoyancy effect.

The demands of relativity allow only three general types of classical field, depending on their behavior under a coordinate transformation. A scalar field is unchanged by a coordinate transformation. A vector field transforms linearly in the transformation matrix, and a tensor field transforms quadratically in the transformation matrix.

The classical fields are called "long range" to distinguish them from the short-range force fields inside the atom. The long-range fields propagate over macroscopic distances. The quantized force bosons of long-range fields are massless. The short-range bosons inside the atom are massive. A long-range vector field has been found in nature: it is the electromagnetic field, and its boson is the spin 1 photon. The gravitational field described

by general relativity is a long-range tensor field, and is presumed to be carried by a massless spin 2 graviton. The graviton has never been detected, and may never be. A long range scalar field would have a massless spin 0 boson, but no long-range scalar field has yet been found in nature. Of the three classical fields, it remains missing. The long-range scalar field should not be confused with the Higgs boson, a scalar field boson recently discovered in accelerator experiments and part of the Standard Model, but the Higgs is massive.

A long-range scalar field then, if it exists, is expected to have a massless boson, such as the photon or the graviton. Yet, like the graviton, the massless boson of the scalar field is expected to go undetected because the scalar field couples weakly to matter. Dicke described how the effects of a long-range scalar field would masquerade as gravity at the classical level, and it would be difficult to distinguish a scalar field from gravity [17,18]. If a long-range scalar field couples to mass, as Dicke assumed, then this scalar field should clothe planetary masses, just as they are clothed with the gravitational field. Yet the energy in the field would be indistinguishable from the mass of the planet it is attached to.

Kaluza proposed a unification of classical gravity, electromagnetism, and a scalar field, by hypothesizing that their respective potentials are all components of a five-dimensional gravitational metric. Ten components of this super-metric are allocated to the standard 4-dimensional (4D) metric of general relativity, four components to the electromagnetic vector potential of the Maxwell equations, and one component to the unnamed scalar field. When the Einstein equations are applied to this 5D metric, one recovers the field equations of general relativity including energy-momentum sources in the electromagnetic and scalar fields, the Maxwell equations in curved spacetime, and a new equation for the scalar field. When the geodesic hypothesis is applied to the metric, one recovers the standard 4D geodesic equation with both the Lorentz electromagnetic force and a new scalar field force.

The classical Kaluza field Lagrangian $L_{\rm K}$ in terms of the gravitational metric $g_{\mu\nu}$ and associated Ricci tensor $R_{\mu\nu}$, the electromagnetic vector potential $A^\mu$ and associated Maxwell tensor $F_{\mu\nu}$, and a scalar field $\phi$, is given by:

$$L_{\rm K} = g^{1/2}\left[\phi g^{\mu\nu}R_{\mu\nu} - \frac{1}{4}\phi^3 g^{\alpha\mu}g^{\beta\nu}F_{\alpha\beta}F_{\mu\nu}\right], \tag{1}$$

where greek indices span the 4 spacetime coordinates, and summation is implied over repeated indices.

The Kaluza scalar field is similar to other scalar field theories, but it is distinguished by its unique coupling to both gravity and electromagnetism. That is what allows us to contemplate a scalar field gravity test using electrostatic equipment.

It should be emphasized that we are treating predictions of a particular species of the Kaluza theory, encapsulated by the Lagrangian above. We are considering specifically (i) no dependence on derivatives with respect to the fifth coordinate, (ii) purely classical theory, (iii) macroscopic fifth dimension. All three of these original features of the Kaluza theory have been relaxed or altered in variations of the Kaluza theory. After the discovery of quantum mechanics, assumptions (ii) and (iii) were widely abandoned in favor of a quantized, microscopic fifth dimension, and the introduction of the Planck constant into the parameterization. Other authors have abandoned assumption (i), which leads to explosive complexity in the theory, yet the retention of assumption (i) elegantly insures that electric charge is an invariant quantity while still adding the complexity of a dynamical scalar field.

The three assumptions, as stated above, form the core of the most conservative species of the Kaluza theory. It generally can be understood as Kaluza's original classical theory, modified by decades of research into the scalar field equations. As can be seen from the Lagrangian $L_{\rm K}$, when the scalar field $\phi = 1$, the theory reduces exactly to classical general relativity and electromagnetism.

This very identity has made the theory difficult to test experimentally, because it has been unclear how to generate the scalar field necessary to effect a departure from known classical physics. The results presented in [14] predicted, for the first time, apparently large

forces in the presence of a scalar field near unity, $\sim 1 \pm 10^{-10}$. That is what we tested in the results reported here.

The scalar force buoyancy mechanism identified in [14] was developed in a linear, Newtonian approximation for spherical symmetry about a mass, such as a planet. As Dicke expected [17,18], a planetary mass will generate a scalar field, and the scalar field will be as weak as gravity in relative terms. Yet a Dicke-style scalar field has no effect in the force equation, and no separate "scalar charge", by assumption. The Kaluza scalar field does have a separate "scalar charge", and does enter the force equations. The author in [14] identified the scalar charge in terms of the electric charge and mass.

The classical Kaluza scalar field sign found by Ref. [14] is opposite to that expected by Dicke. It is rather more similar to the field of an electric charge, described by the Reissner–Nordstrom metric. It still has positive energy and still contributes positively to the gravity of the planet. However, the usual gravitational "well" around a planet is not merely deepened from the scalar field, as Dicke assumed, but rather, there is overlaid a potential "hill" from the scalar field, so that is pushes outward and upward from the planet on bodies carrying scalar charge, just as an electric field would push outward on bodies carrying electric charge. This is shown in Figure 1.

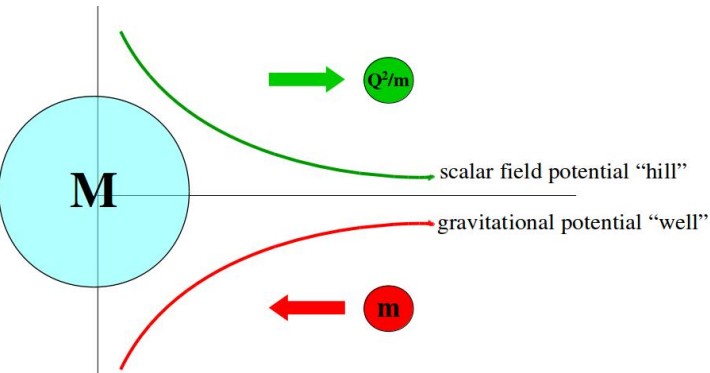

**Figure 1.** Schematic diagram of upward scalar force and downward gravitational forces [14], on a body of mass *m* and electric charge *Q*, from a planet of mass *M*. Both potentials vary as $1/r$, where *r* is distance from the center of the planet. Scalar charge is proportional to $Q^2/m$, so electrically charged objects are predicted to be "buoyant" in the scalar field, contributing an upward force against the downward gravitational force.

The magnitude of the upward force was predicted by [14] to be directly proportional to the weight, similar to a buoyancy force, with a coefficient depending on the dimensionless charge-to-mass ratio parameter described above.

The equation for the gravitational force of a body of mass *m* in the field of a planet of mass *M* clothed with scalar field is given by:

$$\mathbf{F}_{\mathrm{g}} = -mc^2 \nabla \left( \frac{-GM}{rc^2} \right) = mG \nabla \left( \frac{M}{r} \right), \tag{2}$$

where *r* is radial distance from the center of the planet, and *c* is the speed of light.

The equation for the scalar force for a body with electric charge *Q* in the field of a planet of mass *M*, is given in [14]:

$$\mathbf{F}_s = -\frac{c^2 Q^2/m}{16\pi G \epsilon_0} \nabla \left( \frac{GM}{3rc^2} \right) = -\frac{Q^2/m}{48\pi \epsilon_0} \nabla \left( \frac{M}{r} \right). \tag{3}$$

Because the scalar field force and gravitational force both have the same functional dependence, the ratio of upward scalar force to downward gravitational force depends on the charge $Q$ and mass $m$ of the body only, with the mass $M$ of the planet divided out:

$$\mathbf{F}_s = -\frac{Q^2/m^2}{48\pi G\epsilon_0}\mathbf{F}_g. \tag{4}$$

The scalar charge of an object was found to be proportional to the square of the electric charge, and inversely proportional to the mass. The three "charges" of mass, electric charge, and scalar charge, are related in the 5D theory because electric charge is related to motion along the fifth coordinate, and the 5-velocity—comprising energy, momentum, and electric charge—has invariant length.

It is little-appreciated that the strength of gravitational forces at earth's surface arise from a unitless potential that is $10^{-10}$. The field is weak, but the force is strong because the gravitational "charge" is $mc^2$, the energy, which can be large. A similar situation holds for the scalar charge against the weak scalar field, also $\sim 10^{-10}$, but the predicted forces are much stronger than gravity for large electric charge states. The predicted forces can be extremely large for charge states available in the laboratory, and these were not observed.

Equation (4) describes the predicted scalar forces in units of the weight of the test object. The dimensionless ratio in (4) is the square of the parameterization suggested in the previous section, and tabulated in the results of Tables 2 and 3. Although the force predicted from the Kaluza theory scales with the square of this number, we tabulate in this way to allow for intercomparison with other electro-gravitic test designs.

**Table 2.** High voltage series load cell output (mV) test data for acorn and sphere, at each polarity. Unitless electro-gravitic parameter also shown. Load cell calibration factor 0.0016 mV/g. Sphere and plus-polarity acorn data shown are representative of two additional series taken at each configuration. Minus-polarity acorn data shown are representative of one additional data set. All measurements consistent with null effect.

| Voltage (kV) | $\dfrac{Q/M}{\sqrt{4\pi\epsilon_0 G}}$ | Acorn (3 kg) Plus pol. (mV) | Minus pol. (mV) | $\dfrac{Q/M}{\sqrt{4\pi\epsilon_0 G}}$ | Sphere (600 g) Plus pol.(mV) | Minus pol.(mV) |
|---|---|---|---|---|---|---|
| 0 | 0 | 5.286 | 5.347 | 0 | 1.182 | 1.192 |
| 10 | 1000 | 5.285 | | 2300 | 1.182 | 1.192 |
| 20 | 1900 | 5.284 | | 4700 | 1.182 | 1.192 |
| 50 | 4900 | 5.285 | 5.344 | 12,000 | 1.181 | 1.192 |
| 100 | 9700 | 5.285 | 5.346 | 23,000 | 1.182 | 1.193 |
| 150 | 15,000 | 5.286 | 5.346 | 35,000 | 1.182 | 1.191 |
| 200 | 19,000 | | 5.346 | 47,000 | | |
| 0 | 0 | 5.283 | 5.346 | 0 | 1.181 | 1.192 |

All the charge states tested and reported here had dimensionless charge-to-mass ratios greater than unity. Our lowest test value in the dimensionless ratio was the test at 10 V for the 3 kg test article, reported in Table 3. If the prediction reported in [14] were correct, it would correspond to a 100% weight reduction. All the other higher-ratio tests would produce upward forces more than the total weight, beyond our ability to measure. Yet, we measured no discernible weight reduction for any charge state, with our weight sensitivity proven to be 2 g. We had the ability to measure a 100% weight reduction, and such a reduction was predicted by the Kaluza theory for all tests, but we measured zero to within an accuracy of order 0.1% for all tests. The data reported in Tables 2 and 3 are the raw load-sensor voltage readings.

**Table 3.** Moderate voltage series test data for acorn and sphere, at each polarity. Unitless electro-gravitic parameter also shown. Load cell calibration factor 0.0013 mV/g. Data shown are representative of two additional series measured for each configuration. All measurements consistent with null effect.

| Voltage (volts) | $\dfrac{Q/M}{\sqrt{4\pi\epsilon_0 G}}$ | ——— Acorn (3 kg) ——— | | $\dfrac{Q/M}{\sqrt{4\pi\epsilon_0 G}}$ | ——— Sphere (600 g) ——— | |
|---|---|---|---|---|---|---|
| | | Plus pol. (mV) | Minus pol. (mV) | | Plus pol.(mV) | Minus pol.(mV) |
| 0 | 0 | 5.309 | 5.375 | 0 | 1.158 | 1.181 |
| 10 | 1.0 | 5.310 | 5.374 | 2.3 | 1.158 | 1.180 |
| 20 | 2 | 5.311 | 5.374 | 5 | 1.158 | 1.181 |
| 50 | 5 | 5.309 | 5.373 | 12 | 1.158 | 1.180 |
| 100 | 10 | 5.310 | 5.373 | 23 | 1.158 | 1.180 |
| 200 | 20 | 5.307 | 5.373 | 47 | 1.157 | 1.181 |
| 500 | 50 | 5.307 | 5.372 | 120 | 1.158 | 1.180 |
| 1000 | 100 | 5.306 | 5.373 | 230 | 1.159 | 1.180 |
| 2000 | 200 | 5.304 | 5.371 | 470 | 1.159 | 1.180 |
| 5000 | 500 | 5.300 | 5.371 | 1200 | 1.161 | 1.180 |
| 0 | 0 | 5.299 | 5.370 | 0 | 1.161 | 1.180 |

## 3. Test Configuration and Methodology

### 3.1. Overview

Our test configuration measured only components along the vertical direction for any forces generated. This would also test any electric influences on gravity. The general configuration is shown in Figure 2.

Two test articles were used. Both were hollow with conducting surfaces. One was a sphere of mass 600 g and diameter 22 cm. One was an acorn-shaped object of mass 3 kg and average diameter 46 cm, so the charge distributed on its surface will not be uniform, as for the sphere.

The test articles were suspended 3 m from a metal ceiling by a non-conducting nylon cord.

A custom-made load cell that converts the weight to a voltage signal was placed at the suspension point. The electronics to read the cell were remote from it, and shielded from the high voltages.

Three different charging devices were used to charge the test articles. One was a Van de Graaff generator with a realized 200 kV capability, operated at negative polarity. One was a Glassman generator with a 150 kV capability operated at a single polarity. The third was a Kepco moderate-voltage power supply, to provide voltages from 0 to 5 kV at both polarities.

A custom-made high-voltage divider was used to read the high voltages without discharging the test articles.

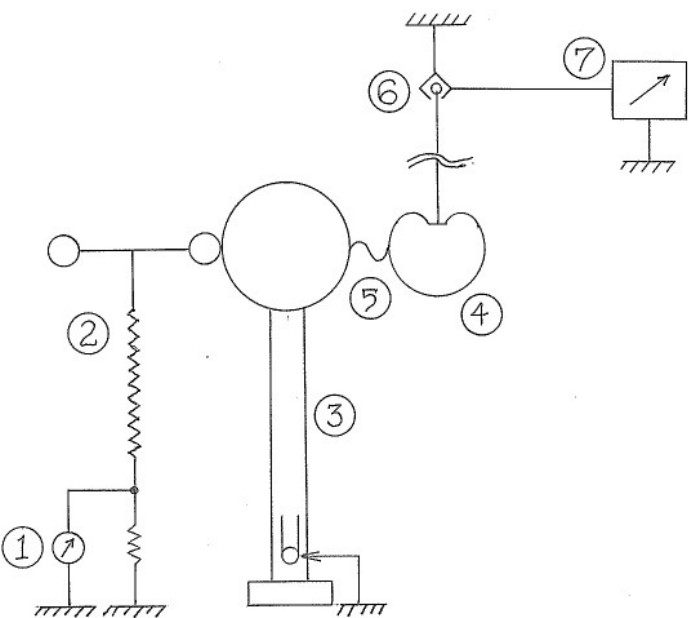

**Figure 2.** Schematic diagram of the experiment design. (**1.**) Electrostatic voltmeter, (**2.**) Custom voltage divider, (**3.**) Van de Graaf (VdG) electrostatic machine, (**4.**)"Acorn" test article suspended from load cell, (**5.**) Flexible conductor between VdG and acorn, (**6.**) Load cell suspended from ceiling, (**7.**) Load cell readout. The test article position remains fixed when the moderate-voltage charge source is used.

Every test monitored the weight change going from zero to the maximum voltage and back down again. This allowed for searching a range of intermediate charge states.

The voltage indicator and control, and the weight output, were in separate locations and monitored simultaneously by two experimentalists. The weight output was continuously monitored, while the voltage was altered.

Calibration measurements were taken on the load cell before, after, and during test runs. A calibration measurement consisted of adding a known weight to the test article and recording the change in voltage from the load cell, and then removing the test weight and recording the change in load cell voltage. Test weights used were 2, 5, and 10 g. The calibration was always recorded as a ratio of voltage to weight, and the load cell response was linear for the tests.

### 3.2. High Voltage Supply

A custom-fabricated Van de Graaff (VdG) electrostatic machine with a theoretical maximum voltage (with respect to ground) of approximately 900 kV was used in this experiment (see Figure 3). In practice, this voltage could never be attained due to various factors, such as humidity, surface roughness, charge leakage, etc. The practical voltage limit of the VdGs was, therefore, approximately 500 kV. When a VdG was connected to a test article, however, the maximum sustained, repeatable voltage on the test article was measured to be 200 kV.

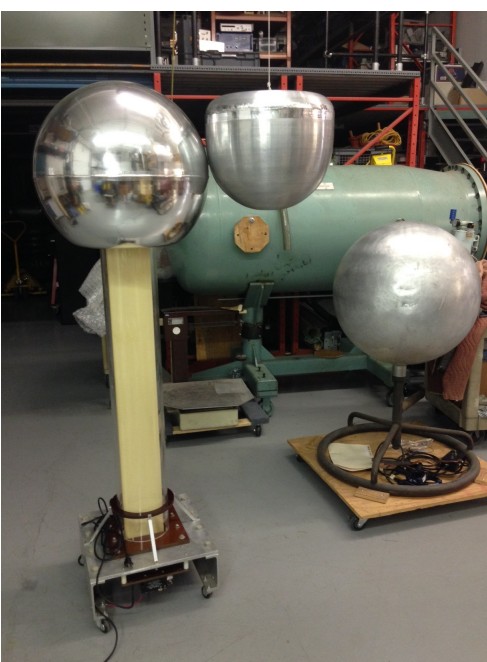

**Figure 3.** Acorn suspended next to a Van de Graaff generator.

The other high-voltage (HV) source was a Glassman model 150R-003 150 kV power supply, operated at a single polarity. This power supply was placed on a movable hydraulic lift platform such that its upper high voltage terminal was at the same height as the VdGs, so that the test articles had the same spatial configuration for all measurements.

The voltage controls for the various voltage supplies were kept remote from the electrified test article.

### 3.3. High Voltage Measurement

Accurate measurement of hundreds of thousands of volts is difficult. A precision, HV divider capable of measuring DC voltages up to 1 MV was used to provide an accurate voltage measurement at high voltages. A voltage divider will reduce the voltage by a known ratio before measurement, allowing better accuracy by measuring at a lower voltage. Standard HV dividers are typically used in HV engineering laboratories where the current is several amperes or kiloamperes. However, since the VdGs are very low current (microampere) devices, commercial HV dividers would present too much of a burden to the VdG, further reducing the maximum voltage possible. The low-burden HV divider used in this test was custom built, with a calibrated divider ratio up to around 300–500 kV, and uncalibrated division up to 1 MV. The calibrated ratio is 50:1, and is accurate to better than 1%. The output of the voltage divider was measured by a precision electrostatic volt meter.

### 3.4. Moderate Voltage Supply

A moderate-voltage power supply was used to explore lower voltages. For these tests we used a Kepco model OPS5000, with a 5 kV capability, modified to allow voltage control by means of a high-voltage insulated 20-turn potentiometer remote from the power supply. This was done to ensure that spurious weight-change signals due to electrostatic or other influences could be minimized. The voltage was monitored using a calibrated 40 kV HV probe connected to a Fluke 87 voltmeter. The 5 kV supply was placed on a movable hydraulic lift platform, to which was attached a wooden "2-by-4" beam about 1 m long. To the top of this beam was attached a high-voltage standoff whose terminal faced the test article but separated from it by a few tens of centimeters. The output high voltage lead (either polarity) of the 5kV power supply was fed to this standoff, to which

was also attached a limp copper braid, the same braid as used in the HV experiments: see Figures 3 and 4. The Kepco voltages are accurate to $\pm 0.5\%$.

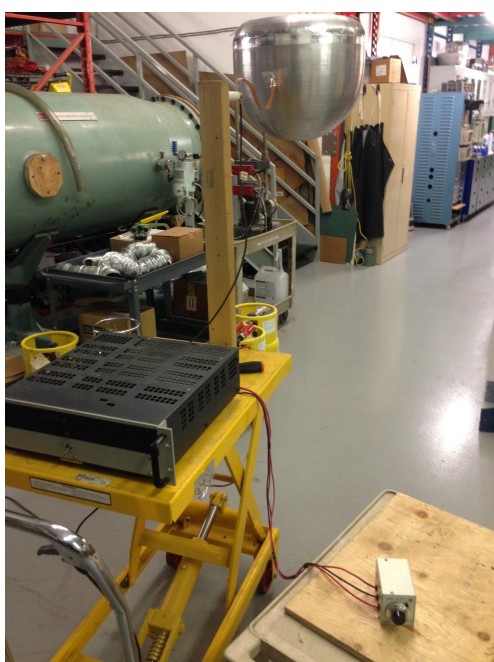

**Figure 4.** Acorn in the low-voltage configuration. The flexible copper braid connecting the power supply to the acorn is visible.

### 3.5. Test Articles

Two test articles were used. Both were hollow with conducting outer surfaces. One article was a sphere of radius about 11 cm and weight approximately 600 g. The other was shaped like an acorn, with an average radius of 23 cm and mass of about 3 kg.

### 3.6. Weight Measurement

The test articles were suspended about 3 m below a metal ceiling, and 2 m above a concrete floor, by a thin nylon cord. Lab walls and extraneous equipment were several meters away. Therefore, the test articles were isolated in a rather large volume in the lab.

A custom-designed load cell was used to measure any weight change of the test articles. The high voltages present necessitated a specially-shielded design, incorporating a 5 kg capacity Wheatstone strain-gauge load cell, capable of 1 g resolution in normal use. In this experiment, however, the load cell was calibrated to 2 g within the expected high-charge regime. The load cell outputs a small DC voltage proportional to the weight, and the voltage reading from the load cell is the measured data. The electrical output read from the load cell was maintained at a large distance from the region of high electrostatic field.

Load cell output calibration was checked constantly before, during, and after the individual tests. For either of the test articles, calibration was accomplished by adding and removing precision test weights of 2.0, 5.0, and 10.0 g to the suspended test article. The difference in voltage on the load cell was recorded for the difference in weight. A calibration measurement was recorded upon the addition of the test weight, and upon its removal. The data of each calibration measurement are the ratio of voltage change to weight change. The calibration factor measured over 9 calibration series in advance of HV testing, 3 with each test weight against the acorn, is $0.0016 +/- 0.0005$ millivolt per gram weight change. For the low voltage test series, the calibration was measured at $0.0013 +/- 0.000$ mV/g.

Therefore, during tests on the acorn, the load cell output voltages measured were in the range of 5 mV for the acorn, and 1 mV for the sphere. Deviations from these baseline values were sought for a range of voltages, with a resolution of about 2 g for either test article.

*3.7. Experiment Configuration*

In order to charge the test article to the desired high voltage, the test article was electrically connected to the power supply so that the charging connection would not unduly influence the weight measurement. A thin flat braid of fine copper wire on cotton was sufficiently limp so as to not influence the weight measurement down to the level of resolution, and this was used to attach the power supply to the test article during the test series.

The voltage divider was directly connected to the Van de Graaff by means of conductive tape, and did not affect the weight measurement. The output of the 50:1 voltage divider was fed to a sensitive electrostatic voltmeter, where each kilovolt reading corresponded to 50 kV on the VdG.

The various power supplies, the voltage divider, the safety grounding rod, and the load cell wiring and mounting box were all attached by copper strap to a local earth ground.

To ensure that the strong electrostatic field was not influencing the load cell measurements, a control test was performed with the acorn mounted on a non-conductive support which took all its weight. Then the VdG was turned on to its fullest charge and the load cell output (weight change) was measured. There was no weight deviation down to the 0.002 mV range, indicating a negligible influence on the measurement from the electrostatic field.

Figure 2 is a diagram of the experimental configuration with a test article in place and a VdG charge supply.

Figure 3 shows the acorn test article suspended next to a VdG.

Figure 4 shows the acorn in the low voltage test configuration.

## 4. Test Results

Twenty-three separate test series were executed, with a series defined as a range of voltages tested for a particular polarity, test article, and power supply. Just two test series were executed above 150 kV. After the test series, the calibration was checked with a 2 g test weight added to the test article.

The negative-polarity VdG was used for the two series above 150 kV, with the acorn test article. Figure 2 shows the general setup. For each test series, the control test using the supported acorn was performed. Then the acorn was freely suspended from the load cell uncharged. By means of varying the belt speed of the VdG, the voltage was slowly raised in increments of 50 kV to a maximum of 200 kV while measuring the weight change. The belt speed and weight monitor were at separate stations, manned by two experimentalists. Upon reaching the maximum voltage for the measurement in the series, the voltage was slowly returned to zero, and the weight was monitored continuously during this time. A complete measurement in the series took several minutes, with a pause at each specified voltage. The high voltage test series data are shown in Table 2. Only one of these series is shown in Table 2, since both series were similar.

The other 21 series tested 7 different parameter configurations, with 3 series for each configuration. The 7 configurations were high-voltage testing (Glassman) of the acorn at positive polarity, the sphere at high voltage at each polarity, the acorn at low voltage (Kepco) at each polarity, and the sphere at low voltage at each polarity. In the data presented below, just 1 of the 3 series for each of the 7 configurations is shown, since the variation between sets was negligible. Note that the high-voltage test results with Glassman and VdG power supplies are combined in Table 1, since we only distinguish voltage value and not source.

The test procedure was to increase the voltage from zero to maximum, pausing for several seconds at each voltage, then slowly increasing the voltage to the next level, and so on. After the maximum voltage in the series was reached, the voltage was decreased slowly back to zero, while continuously monitoring the weight. A zero-charge reading was taken at the end of each series, and in some cases, the zero-charge weight drifted over the course of the series. At various times during the measurements, a calibration check was

done to ensure that the load cell was still reading correctly. The moderate-voltage test data are shown in Table 3.

### 5. Conclusions

All the charge states tested and reported here had dimensionless charge-to-mass ratios greater than unity. Our lowest test value in the dimensionless ratio was the test at 10 V for the 3 kg test article, reported in Table 3. If the prediction reported in [14] was correct, it would correspond to a 100% weight reduction. All the other, higher-ratio tests would produce upward forces of more than the total weight, yet we made no attempt to measure net upward forces. We measured no discernible weight reduction for any charge state, with our weight sensitivity proven to be 2 g. We had the ability to measure a 100% weight reduction, and such a reduction was predicted by the Kaluza theory for all tests, but we measured zero to within an accuracy of order 0.1% for all tests. Therefore, we consider our tests to falsify the classical Kaluza prediction of strong electro-gravitic buoyancy forces from electrostatic coupling to the scalar field of the earth.

This appears to be the first experimental falsification of the classical Kaluza theory, for a macroscopic fifth dimension and a scalar field near unity. The Kaluza theory becomes identical with standard electromagnetism and gravity in the limit of weak scalar fields, so this is the first test of the classical theory in such weak-field regimes.

Taken as a generic measurement irrespective of theoretical expectation, measurements were consistent with a null effect to within a 2-gram precision. Drifts were seen in the load cell up to 0.01 mV over a test series, but the zero measurement at the end of the series confirmed the drift was a measurement artifact. The load cell uncharged reading drifted by as much as 0.025 mV between series, a magnitude similar to the resolution measured at calibration. Yet a measurement at zero charge always showed the drift was not a real effect of the electric charge.

We attribute this drift in load cell reading of the zero value during a test series as due to a temperature dependence of the load cell elements. Near the low end of the measurement range, a small temperature fluctuation can affect the load cell output at the order of magnitude seen in these drifts. Since this was a large-volume experiment it was difficult to control ambient temperature to within several degrees, and this was sufficient to produce drift in the zero measurement during the course of the series.

Further work might search for effects in the parameter regime characterized by fractional electro-gravitic ratios; the smallest ratio considered here was 1.0 and the largest 47,000. Our results provide good logarithmic coverage from $10^1$ to $10^4$. It might be profitable to investigate small electro-gravitic ratios, from 1 to 0.001, for example.

**Author Contributions:** Conceptualization, L.L.W.; methodology, G.H. and L.L.W.; validation, G.H. and L.L.W.; formal analysis, L.L.W.; investigation, G.H.; resources, G.H.; data curation, G.H.; writing—original draft preparation, L.L.W.; writing—review and editing, L.L.W. and G.H.; visualization, G.H. and L.L.W.; supervision, L.L.W.; project administration, L.L.W; funding acquisition, L.L.W. All authors have read and agreed to the published version of the manuscript.

**Funding:** This research was funded by DARPA DSO under award number D19AC00020.

**Data Availability Statement:** The data reported in this study are included in Tables in the main article.

**Conflicts of Interest:** The authors declare no conflict of interest.

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
