# Peer review of "Null-Result Test for Effect on Weight from Large Electrostatic Charge"

_2624-8174, doi:10.3390/physics3020013_

Round 1
Reviewer 1 Report
In this work, the authors had designed and carried out an experiment to measure and verify the predicted effects that caused by certain coupling between gravitation and electromagnetism mediated by a suggested classical long-range scalar field. Even though the resolution of this experiment may be far from the magnitudes of the expected signals, the null results may still serve as an independent constraint of certain theoretical models.
But, I believe, at this stage, the experiment and the conclusion of this work are far from convincible. According to the standards of this Journal, I suggest that, before the reconsideration of this work, the authors should answer the following commons and make revisions.
The authors need to explain and describe clearly the theoretical model under considerations, and the related parameters that the authors tried to measure in their experiments. Especially, they should explain how the expected weight variations of the gradually charged object depends on the detailed models, say the coupling model involving a long-range scalar field or the classical Kaluza model mentioned in the manuscript, and how such weight variations, if exist, can be distinguished from the rest mass variations predicted by the special theory of relativity. The large buoyancy force mentioned in the manuscript that found by one of the authors is not of a “common sense” in the literature, the authors need to explain the prediction and the relations with the experiment. In other words, the authors should explain why such an experiment of weight measurements could be employed to set constraint on the fundamental coupling models of electromagnetism and gravitation.
The authors should then estimate the magnitudes of the expected signals, according to their experiments setup and the theoretical model explained as required. I believe the readers need to understand how an experiment with resolutions about 1 g and test object about 3 kg could be used to test such relativistic effect. This may confuse the readers because we generally do not expect such large mass variations of a test object involving interactions under such low energy scale.
At last, the classical Kaluza theory or general relativity in higher dimensional spacetime had already been tested by many experiments. To emphasize that their experiment is the first test of the classical Kaluza theory in the weak field regime is not appropriate.
Author Response
Response:
Thank you to the reviewer for the time and consideration. In summary, the reviewer asks for more detail on the theoretical justification for the Kaluza mechanism, and how our experimental results quantitatively constrain that theory. We are happy to provide that detail, and to also address the other concerns of the referee.
In broad summary, there is a new section 2 "Kaluza Buoyancy Force Detail", that addresses the requests of the referees. The updated PDF is attached below. There is also a new Figure 1 that illustrates the scalar force interaction being tested in our experiments.
We tried to develop new descriptive content for the Kaluza buoyancy effect, including the Figure 1, without merely copying from the theory reference. Yet the theory paper Ref.12 is available online by a reputable open-access publisher. It is peer-reviewed, archived, and indexed, so it should be sufficient to reference the details in that paper. Therefore we only carried over the key force equations from Ref.12.
The conclusions have also been modified also to address the reviewer concern.
Now let us address the specific comments:
The authors need to explain and describe clearly the theoretical model under considerations, and the related parameters that the authors tried to measure in their experiments.
Response: we added section 2 to this purpose. It summarizes and illustrates the nature of the Kaluza effect we tested. However, we still recognize that our measurement may be of interest independent of the Kaluza effect, and related and similar experiments are parameterized in Table 1.
Especially, they should explain how the expected weight variations of the gradually charged object depends on the detailed models, say the coupling model involving a long-range scalar field or the classical Kaluza model mentioned in the manuscript,
Response: section 2 contains the precise parameterization from the calculation of Ref.12, which is published at a reputable open-access, peer-reviewed journal. We do not wish to repeat content already published, but we hope the detail we did provide is sufficient to fully explain the physical content of the theory and experiment.
and how such weight variations, if exist, can be distinguished from the rest mass variations predicted by the special theory of relativity.
Response: Our apologies, but we could not understand this short comment. Our tests are static tests, with all objects at rest. There should be no special relativistic mass variation expected from our experiment. Please elucidate if we have misunderstood.
The large buoyancy force mentioned in the manuscript that found by one of the authors is not of a “common sense” in the literature, the authors need to explain the prediction and the relations with the experiment. In other words, the authors should explain why such an experiment of weight measurements could be employed to set constraint on the fundamental coupling models of electromagnetism and gravitation.
Response: Section 2 addresses these concerns. We discuss explicitly why it is like a buoyancy. We hope this description is satisfactory.
The authors should then estimate the magnitudes of the expected signals, according to their experiments setup and the theoretical model explained as required. I believe the readers need to understand how an experiment with resolutions about 1 g and test object about 3 kg could be used to test such relativistic effect.
Response: Section 2 also quantifies the effect relative to gravity, and discusses the resolution of our experiments. We agree wholeheartedly with the sentiment of the comment and appreciate the opportunity to improve our content in that area.
This may confuse the readers because we generally do not expect such large mass variations of a test object involving interactions under such low energy scale.
Response: We tried to also in section 2 clarify that this is a separate force, from a separate force field. It is not a variation in mass. It is an upward force, like a buoyancy force. We are grateful for the comment, and we hope we have sufficiently clarified this important point. Figure 1 we hope makes it clear.
At last, the classical Kaluza theory or general relativity in higher dimensional spacetime had already been tested by many experiments. To emphasize that their experiment is the first test of the classical Kaluza theory in the weak field regime is not appropriate.
Response: In section 2, we attempted to make clear that our test is particular to one "species" of the Kaluza theory, and we enumerated 3 properties. The classical Kaluza theory we investigated, for a macroscopic fifth dimension and under the cylinder condition, has not been verified previously in the manner we did. We do believe our statement in the abstract is true, "this is the first unipolar, high-voltage, meter-scale, static test for electro-gravitic effects reported in the literature". There is a broad range of variations on the Kaluza theory and invoking microscopic dimensions and any number of quantum phenomena, but that is different than the classical, macroscopic unified theory of gravity and electromagnetism that we are investigating. It is also the first report of such results for any theoretical reason. We please request the reviewer provide any references and we will gladly include them. We apologize if our literature search has been lacking, but the dearth of such experimental results is another thing that we think makes our results of interest.
Thank you for your consideration.
George Hathaway
Lance Williams

Reviewer 2 Report
The paper discusses experimental tests of the Kaluza-Klein model unifying gravity and electromagnetism. As it is known, the unified five-dimensional Kaluza-Klein-like theory involves four-dimensional gravity,. electromagnetic and scalar fields. Within this paper, the authors discuss experiments aimed to keep track of the scalar field and find that if scalar field is weak (which is natural), and no predicted large buoyancy force is detected.
While the experiment is very interesting and discussed in great details, and the conclusion that actually there is no unusual buoyancy forces is clearly very important, the discussion of theoretical base is very superficial. So by my opinion, to be published, the paper should involve a better theoretical discussion of motivation, explanation of buoyancy forces and a some brief review on Kaluza-Klein theory to help a reader to understand the concept of idea tested by the experiment.
Author Response
Response:
Thank you to the reviewer for the time and consideration. In summary, the reviewer asks for more detail on the theoretical justification for the Kaluza mechanism, and how our experimental results quantitatively constrain that theory. We are happy to provide that detail, and to also address the other concerns of the referee.
In broad summary, there is a new section 2 "Kaluza Buoyancy Force Detail", that addresses the requests of the referees. There is also a new Figure 1 that illustrates the scalar force interaction being tested in our experiments. An updated PDF is attached below.
We tried to develop new descriptive content for the Kaluza buoyancy effect, including the Figure 1, without merely copying from the theory reference. Yet the theory paper Ref.12 is available online by a reputable open-access publisher. It is peer-reviewed, archived, and indexed, so it should be sufficient to reference the details in that paper. Therefore we only carried over the key force equations from Ref.12.
To the reviewer specific comment:
While the experiment is very interesting and discussed in great details, and the conclusion that actually there is no unusual buoyancy forces is clearly very important, the discussion of theoretical base is very superficial. So by my opinion, to be published, the paper should involve a better theoretical discussion of motivation, explanation of buoyancy forces and a some brief review on Kaluza-Klein theory to help a reader to understand the concept of idea tested by the experiment.
Response: We believe the material in section 2 will satisfy the reviewer request for additional information on the buoyancy force. We are grateful for the comment and the opportunity to add additional detail on this interesting effect. But we still feel our results may have general interest beyond the Kaluza theory, since these tests are the first such ever reported in the literature.
Thank you for your consideration.
George Hathaway
Lance Williams

Round 2
Reviewer 2 Report
By my opinion, the authors properly addressed my suggestions and comments, so, now the paper can be published.